## [Transparent Peer Review file · Nature Communications]

Organic di-selenide hydrogel microspheres for multimodal treatment of osteoarthritis

Corresponding Author: Professor Xuesong Zhu

Version 0:

Reviewer comments:

Reviewer #1

(Remarks to the Author)

The technology reported here is interesting, although the novelty is difficult to evaluate. In this manuscript, the authors constructed a ROS/MMP13 dual-responsive organoselenium hydrogel microsphere (HSPHR), which can significantly upregulate the expression of TXNRD1, activate the PI3K-AKT-mTOR signaling pathway, and enhance mitophagy and mitochondrial function. The aim is to promote cartilage repair and functional reconstruction through multidimensional targeted intervention. We appreciate the authors' efforts; however, there are some questions and concerns that arose during the review of the manuscript, which are listed as follows:

1. The authors have not provided evidence to confirm the successful preparation of Se NPs as opposed to other possible substances. Notably, they conducted both in vitro and in vivo studies using these materials without first performing any material characterization experiments to verify the identity and properties of the Se NPs.
2. The font size in Figures 4I-J is too small, which hinders effective extraction of information from the images. It is recommended that the authors increase the font size to improve readability and clarity.
3. On page 18, line 7, the authors state that "HSPHR exhibited accelerated degradation when exposed to both H₂O₂ and the MMP13 enzyme." However, in DMM mice, the degradation rate of HSPHR appears to be slower.
4. In Figure 6A, the specificity of the PI3K protein bands appears to be suboptimal. Furthermore, the consistency of the AKT and P-AKT results raises doubts as to whether these proteins were detected on the same membrane. Please provide the Western data with whole membrane and shown with marker ladder.
5. In Figures 5–6, a group without IL-1 β treatment should be included as a blank control to verify the successful induction of inflammation by IL-1 β . The absence of such a control group makes it difficult to determine whether the observed effects are specifically due to IL-1 β -induced inflammation.
6. Several aspects of Figure 7 require clarification:
 - a. The S.O. staining of cartilage in the DMM group appears to be more intense, whereas the staining in the HSPHR group is unexpectedly lighter. The authors should explain the reasons for these observations.
 - b. The immunohistochemical staining results for TXNRD1 and GPX1 in the DMM group are inconsistent with those of other groups, with marked differences in nuclear staining intensity. This raises concerns about the experimental procedures and comparability among groups.
 - c. Previous studies (Bone Res. 2019 Aug 5;7:23. doi: 10.1038/s41413-019-0062-y) have demonstrated that AKT signaling is activated during OA. However, in this study, the level of p-AKT in DMM mice is significantly decreased. The authors should provide a rationale for this discrepancy.
 - d. The number of biological replicates per group (n=4) is insufficient to provide robust evidence that HSPHR alleviates OA progression.
7. The naming of the control groups in Figure 6D is inconsistent between the representative images and the corresponding quantitative data.
8. On page 21, line 8, the authors mention that "HSPHR hydrogel microspheres improve cartilage metabolic...". However, the manuscript does not provide sufficient evidence to clarify whether TXNRD1 restores mitochondrial function via the PI3K-AKT-mTOR pathway, and the causality between these factors is ambiguous. To strengthen the mechanistic interpretation, the authors should consider assessing mitochondrial function and/or the expression of relevant proteins following activation or inhibition of the PI3K-AKT-mTOR pathway. This would provide more direct evidence for the involvement of this signaling axis in the observed effects.

Reviewer #2

(Remarks to the Author)

The manuscript presents a novel synthesis of a ROS/MMP13 dual-responsive organic double selenium hydrogel microsphere to realize early OA local microenvironmental response by utilizing the pathological characteristics of osteoarthritis, and at the same time, the organic selenium can effectively supplement the selenoprotein level in the cartilage, synovium and subchondral bone. Intra-articular injection of HSPHR not only reduced cartilage degeneration, improved synovial hyperplasia, and delayed sclerosis of the subchondral bone in the post-traumatic OA model, but also accelerated the formation of new cartilage in the cartilage defect model. While the study is innovative and holds significant potential, I would like to provide the following constructive suggestions to enhance the scientific rigor and presentation clarity of this manuscript before it can be considered for publication.

1. The "Introduction" section is too long. Authors should focus only on the most important aspects for a good understanding of the developed work.

2. The authors propose that their "three birds with one stone" modified organic diselenide hydrogel microspheres can simultaneously treat multiple aspects of osteoarthritis. However, this conceptual framework requires clarification for several reasons:

We recommend either:

- a) Providing direct experimental evidence for triple-targeting efficacy, or
- b) Modifying the conceptual framework to better reflect the actual therapeutic mechanisms observed.

This would significantly strengthen the manuscript's mechanistic claims while maintaining scientific rigor. The current presentation risks overinterpretation of the technology's targeting capabilities.

3. While the abstract states that organic selenium supplementation restores selenoprotein levels across multiple joint tissues (cartilage, synovium, and subchondral bone), the study primarily demonstrates mechanistic effects in cartilage without providing selenium quantification data for synovium or subchondral bone. This limitation should be acknowledged, as it affects the interpretation of the compound's proposed multi-tissue targeting capability.

4. While the authors showed HSPHR's inhibitory effect on osteoclast formation in vitro in Supplementary Figure 6, the in vivo study only evaluated late-stage OA pathology (8 weeks post-operation). Since osteoclasts are known to mediate early-stage OA pathogenesis, could the authors address why osteoclast activity was not monitored at earlier time points? This information would strengthen the mechanistic connection between HSPHR's cellular effects and its therapeutic outcomes.

5. The Graphic Abstract does not clearly convey the "three-in-one" concept, nor does it show the microspheres' impact on other cells.

6. To further explore the advantageous effects of HSPHR on cartilage regeneration, the authors developed an in vivo model featuring a full-thickness cartilage defect. Chondrogenic progenitor cells were incorporated into microspheres of HSPHR in order to enhance the therapeutic outcomes for cartilage repair. The study primarily elucidates the role of selenoproteins (particularly GPX1 and TXNRD1) in pathological cartilage, as demonstrated in both the main text and schematic diagrams. However, the relationship between selenoproteins and chondrocyte differentiation remains unclear. Several critical questions need to be addressed:

- a) Molecular Mechanisms: Are GPX1 and TXNRD1 similarly involved in chondrocyte differentiation and cartilage homeostasis? What specific selenoproteins regulate chondrogenic differentiation pathways?
- b) Experimental Evidence: The study would benefit from in vitro experiments using:
 - 1) Chondrocyte differentiation models;
 - 2) Selenoprotein-specific knockdown/overexpression systems;
 - 3) Comprehensive analysis of redox regulation during differentiation.

7. In Figure 4, the authors report solid-state NMR characterization of HA-Se-Se, but the presented spectrum shows only a single peak with no observable selenium signal. This raises several technical and interpretational concerns that need to be addressed.

8. While the manuscript investigates selenium-based therapy for OA, several critical concerns regarding its novelty and mechanistic depth need to be addressed:

Innovation Gap:

The therapeutic use of selenium nanoparticles in OA has been extensively reported (e.g., studies by DOI: 10.1016/j.mtbio.2023.100864; DOI: 10.1016/j.biomaterials.2025.123195; DOI: 10.1016/j.jcis.2025.137612).

The current work must better differentiate itself by:

Selenoprotein Analysis Limitations:

The cartilage selenoprotein expression profiling GPX1/TXNRD1 largely replicates prior work without advancing the field.

Mechanistic Shortcomings:

Which specific selenoproteins mediate the therapeutic effects in cartilage;

How their regulation differs across joint tissues (synovium vs. cartilage vs. bone);

The downstream pathways linking selenoproteins to chondroprotection.

Reviewer #3

(Remarks to the Author)

Comments:

This study developed ROS/MMP13-responsive selenium microspheres to target OA pathology. HSPHR restores selenoproteins and activates PI3K-AKT-mTOR-mediated mitophagy, mitigating cartilage degeneration, synovitis, and subchondral sclerosis in post-traumatic OA models while enhancing regeneration in cartilage defects. While the work addresses an important clinical issue, it falls short in several critical areas, particularly regarding novelty and experimental depth. As such, the current version of the manuscript is not suitable for publication in Nature Communications.

1. The abstract and the introduction should be rewritten, the novelty of the work is not highlighted. What does "Three birds with one stone" mean in the title? It needs to be clearly stated in the introduction.
2. The graphical abstract fails to effectively convey the "Three birds with one stone" concept and requires redesign to explicitly highlight the innovative therapeutic strategy of the HSPHR system.
3. The article uses a great many abbreviations, some of which are not explained at all (e.g. CEMIP, HSR, DMM, Ser, PTEN, WB, etc.). The abbreviations of all text are very confusing. Full names should be given when an abbreviation first appears in the text. Additionally, the manuscript contains redundant abbreviation definitions (e.g., osteoarthritis (OA) appears with parenthetical clarification multiple times) and inconsistent terminology for key abbreviations. Specifically, the term HSPHR is ambiguously defined as both:
"dual-responsive organicdouble selenium hydrogel microsphere (HSPHR)"
"dual-responsive selenium nanoparticle hydrogel microspheres (HSPHR) "
4. Given that most prior studies refer to "SeNPs" as inorganic selenium nanoparticles, while this work focuses on organic selenium, what specific form does "SeNPs" denote in this context? Please discuss the advantages of organic selenium compared to inorganic selenium.
5. The authors claim SeNPs reverse OA cartilage degeneration by restoring antioxidant pathways. However, given the significant influence of nanoparticle morphology and size on bioactivity, the current characterization of these parameters is insufficient and requires enhancement. For instance, comprehensive characterization techniques such as TEM (Transmission Electron Microscopy) for morphology visualization, DLS (Dynamic Light Scattering) for size distribution analysis, and zeta potential measurement for surface charge evaluation should be employed to provide a more accurate and detailed understanding of the SeNPs' physicochemical properties. Additionally, justification is needed for selecting 20 nM and 50 nM concentrations in validating SeNPs' effects on chondrocyte matrix metabolic homeostasis.
6. Please standardize the capitalization of the first letters of the x- and y-axis labels, as well as the font style and size in all figures, and define the abbreviations for each group.
7. The presentation of many data in the manuscript is not standard. All morphological images must include scale bars, such as Figures 2A,3E-M, 5A-D, 7B-M, S7E-I, and S8. All fluorescently stained images should provide information on the fluorescent probes used, such as Figures 5A-D, 7L-M, S6B, and S7E-F, etc.
8. Selenium plays a critical role in maintaining redox homeostasis by participating in antioxidant processes and regulating selenoprotein synthesis. However, this study lacks experimental evidence demonstrating how SeNPs modulate redox homeostasis.
9. The western blot results in Figure 1D, and 6B lack reliability due to inconsistent loading controls.
10. The therapeutic efficacy evaluation of HSPHR microspheres in early-stage osteoarthritis lacks critical validation through imaging modalities (e.g., Micro-CT, MRI) and behavioral assessments (e.g., gait analysis), undermining the clinical relevance of reported histological improvements.
11. Figures 7L-M, S2, S5, S6A-B, S7E, and S9 exhibit poor quality requiring enhanced resolution and annotation. Critically, Figure S5 lacks organ labeling in HSPHR biodistribution images. The significant hepatic retention (at 14 days) raise toxicity concerns, demanding expanded safety studies (hepatorenal function, long-term clearance). These revisions are essential to validate therapeutic biosafety.
12. The discussion section is superficial and lacks depth. The authors should provide a more thorough analysis of their results, comparing their findings with existing literature and highlighting the implications of their work for clinical applications.
13. This manuscript needs a revision. The writing of this paper is too rough. The manuscript requires thorough language revision due to frequent formatting errors (e.g., incorrect subscript/superscript notation in chemical formulas like Na_2SeO_3 and H_2O_2), spelling mistakes (such as ofor, and SeNPss, etc.), and grammatical inconsistencies. Comprehensive English editing is recommended to ensure scientific rigor and clarity.

Version 1:

Reviewer comments:

Reviewer #1

(Remarks to the Author)

The authors have satisfactorily addressed most of my concerns. However, the term "three birds with one stone", while catchy, may still be somewhat overstated without direct comparative efficacy data against other multi-target therapies. The authors should tone down the phrasing unless they can provide comparative evidence.

Reviewer #3

(Remarks to the Author)

Comments: The authors have satisfactorily addressed the majority of my previous concerns through extensive revisions. The

manuscript has been significantly improved in terms of experimental justification, data presentation, and clarity. In my view, the paper now meets the scientific standards of Nature Communications. I recommend acceptance pending minor revisions, primarily focused on further polishing of the language to enhance readability and precision.

Reviewer #4

(Remarks to the Author)

The authors have made substantial improvements in response to the previous reviewers' comments. The manuscript now presents a more complete and coherent study, with clarified methodologies and strengthened data interpretation. The conceptual framework of a ROS/MMP13 dual-responsive organic selenium hydrogel microsphere (HSPHR) for multi-tissue synergistic therapy in osteoarthritis is both innovative and well-supported by experimental evidence. The mechanistic insights into TXNRD1 as a key regulator across cartilage, synovium, and subchondral bone pathways significantly enhance the scientific value of this work.

However, a few minor revisions are still required to meet the journal's publication standards.

Minor Comments:

1. Some subheadings are quite long, for example, "HSPHR hydrogel microspheres improve cartilage metabolic homeostasis in osteoarthritis by restoring mitochondrial function...". It is suggested to refine them appropriately to conform to the standard scientific writing format, which will facilitate readers in quickly locating the key points of the research.
 2. Please provide the full form when an abbreviation appears for the first time (e.g., MRP), and unify the font and line spacing.
 3. Please check whether the label "Glucose" on the ECAR experiment graph in the article should be replaced with "Rot/AA" and make the necessary changes.
 4. The discussion section is overly lengthy and should be streamlined, such as the discussion on SeNPs in Fig. 3.
- Once these minor issues are addressed, the manuscript should be suitable for publication.

Response to Reviewers' Comment

Reviewer #1 (Remarks to the Author):

The technology reported here is interesting, although the novelty is difficult to evaluate. In this manuscript, the authors constructed a ROS/MMP13 dual-responsive organoselenium hydrogel microsphere (HSPHR), which can significantly upregulate the expression of TXNRD1, activate the PI3K-AKT-mTOR signaling pathway, and enhance mitophagy and mitochondrial function. The aim is to promote cartilage repair and functional reconstruction through multidimensional targeted intervention. We appreciate the authors' efforts; however, there are some questions and concerns that arose during the review of the manuscript, which are listed as follows:

Response : We sincerely appreciate the reviewer's constructive suggestions, which have significantly enhanced the quality of our manuscript (see details below).

***Comment 1.** The authors have not provided evidence to confirm the successful preparation of SeNPs as opposed to other possible substances. Notably, they conducted both in vitro and in vivo studies using these materials without first performing any material characterization experiments to verify the identity and properties of the SeNPs.*

Response: We sincerely appreciate the reviewers' valuable comments. The characterization of biomaterials constitutes a critical aspect of our research. SeNPs were comprehensively characterized in our previous studies (*J Nanobiotechnology* 2024;22(1):634), and we apologize for any confusion that may have arisen from the omission of this information in the manuscript. Following the reviewers' suggestions, we have provided a detailed characterization of the SeNPs prepared in this study. Transmission electron microscopy (TEM) was employed to examine the morphological structure of the SeNPs (Figure R1A). The particle size was determined to be approximately 250 nm, which is consistent with the results from dynamic light scattering analysis (Figure R1C). Energy-dispersive X-ray spectroscopy (EDX) confirmed the uniform surface distribution of selenium on the SeNPs (Figure R1B). Additionally, the zeta potential of the SeNPs was measured at -11.7 mV (Figure R1D).

Figure R1. Characterization of SeNPs. (A) Transmission electron microscopy image of SeNPs, (B) Surface EDX spectrum of SeNPs, (C) Particle size distribution of SeNPs, (D) Zeta potential of SeNPs.

Comment 2. The font size in Figures 4I-J is too small, which hinders effective extraction of information from the images. It is recommended that the authors increase the font size to improve readability and clarity.

Response: We sincerely appreciate your valuable feedback. We have carefully corrected all instances of non-compliant font usage throughout the manuscript, including in the main text, and have made comprehensive adjustments in other sections to enhance overall readability.

Comment 3. On page 18, line 7, the authors state that "HSPHR exhibited accelerated degradation when exposed to both H₂O₂ and the MMP13 enzyme." However, in DMM mice, the degradation rate of HSPHR appears to be slower.

Response: We sincerely appreciate your valuable feedback. Through *in vitro* experiments, we have confirmed that HSPHR microspheres exhibit responsiveness to both MMP13 enzyme and reactive oxygen species (ROS), characterized by increased selenium release. To observe this phenomenon in rats, Cy5.5 dye labeling combined with infrared imaging is necessary. Notably, HSPHR contains a selenium-selenium double bond cross-linking structure that facilitates three-dimensional network formation, thereby enhancing drug encapsulation capacity. For instance, previous studies have utilized this selenium-selenium structure to deliver NGF-responsive drugs to brain injury sites for the treatment of Alzheimer's disease (*J Mater Chem B* 2021;9(37):7835-7847). When Cy5.5 is encapsulated within HSPHR, the fluorophore may remain shielded under non-inflammatory conditions due to the stability of

the Se-Se crosslinks. Therefore, fluorescence intensity must be interpreted across two distinct time windows: during the early phase (Day 1–7), abundant fluorophores allow gradual Cy5.5 release, resulting in strong fluorescence in both Sham and DMM groups. During the late phase (Day 8–14), the DMM group exhibits sustained inflammatory-like responses, including elevated ROS and MMP13 levels, which trigger the cleavage of Se-Se bonds and subsequent re-exposure of the Cy5.5 fluorescent moiety. Consequently, the DMM group demonstrates significantly higher fluorescence intensity compared to the Sham group. Similar results have been reported in ROS-responsive delivery systems used in models of middle cerebral artery occlusion (*Acta Pharm Sin B* 2023;13(12):5107-5120) and liver fibrosis (*Biomaterials* 2025;314:122887).

Comment 4. In Figure 6A, the specificity of the PI3K protein bands appears to be suboptimal. Furthermore, the consistency of the AKT and P-AKT results raises doubts as to whether these proteins were detected on the same membrane. Please provide the Western data with whole membrane and shown with marker ladder.

Response: We sincerely appreciate the reviewer's comment. The original blots for all Western blot protein bands have been provided in Supplementary File 1.

Comment 5. In Figures 5–6, a group without IL-1 β treatment should be included as a blank control to verify the successful induction of inflammation by IL-1 β . The absence of such a control group makes it difficult to determine whether the observed effects are specifically due to IL-1 β -induced inflammation.

Response: We sincerely appreciate your insightful suggestions. Implementing blank controls as a quality control measure is essential to ensure the reliability and validity of our experimental results. In response, we have revised the experiments shown in Figures 5–6 (Figures R2–R3) by incorporating blank controls. Our results consistently show that IL-1 β treatment induces an arthritis-associated phenotype in chondrocytes, characterized by reduced COL2 protein synthesis and increased MMP13 expression. Administration of SeNPs and HSPHR effectively mitigates this pathological phenotype (Figures R2A–C), with HSPHR demonstrating a more significant protective effect than SeNPs. Further analysis of

selenoprotein expression revealed that IL-1 β treatment markedly decreases the levels of selenoprotein synthase, GPX1, and TXNRD1. However, treatment with both SeNPs and HSPHR successfully restores the expression of these selenoproteins, with HSPHR showing superior therapeutic potential (Figures R2A, D–F).

Figure R2. Differences in HAMA, 50 ng/mL SeNPs, and HSPHR protein expression following IL-1 β intervention. (A–F) Western blot bands for MMP13, COL2A, GPX1, TXNRD1, and SEPHS1 and their quantitative statistical analysis. (n = 3 per group, *: * P < 0.05, ** P < 0.01, *** P < 0.001, and **** P < 0.001 by one-way ANOVA with Tukey’s multiple comparison test.)

In Figure R3, blank controls were incorporated, and all experiments—including Western blotting, immunofluorescence, and transmission electron microscopy—were reorganized. The addition of IL-1 β significantly downregulated TXNRD1 expression and inhibited phosphorylation of PI3K, AKT, and mTOR. Following HSPHR treatment, TXNRD1 expression was restored to normal levels, thereby reactivating the downstream PI3K-AKT-mTOR signaling pathway. However, administration of the TXNRD1 chemical inhibitor TRi-1 once again suppressed the PI3K-AKT-mTOR signaling pathway (Figure R3A), suggesting that HSPHR exerts its effects through activation of TXNRD1, which subsequently activates the downstream PI3K-AKT-mTOR signaling pathway. Further analysis of mitochondrial structural protein expression, morphology, and functional changes (Figure R3B, G–J) revealed that IL-1 β significantly reduced the expression of mitochondrial intrinsic proteins ATP5A, MT-ND4, COX 4, and TOMM20. HSPHR treatment restored these proteins to baseline levels, whereas TRi-1 administration led to a subsequent decrease in their expression. Similar trends were observed in mitochondrial membrane potential (Figure R3D)

and mitoTracker staining intensity (Figure R3E). Transmission electron microscopy demonstrated that IL-1 β induced mitochondrial swelling and loss of cristae, alterations that were reversed by HSPHR treatment but blocked by TRi-1. Moreover, HSPHR restored IL-1 β -induced reductions in matrix synthesis proteins and suppressed matrix degradation proteins in chondrocytes. Our findings indicate that HSPHR exerts its protective effects by upregulating TXNRD1, thereby activating the PI3K-AKT-mTOR signaling pathway and restoring mitochondrial function and chondrocyte metabolism.

Figure R3. Mechanistic exploration of HSPHR microspheres facilitating selenoprotein synthesis to ameliorate osteoarthritis progression. (A) WB bands of TXNRD1, PI3K, P-PI3K, AKT, P-AKT, and P-mTOR. (B) WB bands of ATP5A, MT-ND4, COX 4, TOMM20, and LC3B. (C) WB bands of ACAN, COL2, MMP13, and ADAMTS5. (D) JC-1 staining. (E) Mitotracker staining and skeletal analysis. (F) TEM of mitochondria. (G) TOMM20 immunofluorescence staining. (H) ATP5A immunofluorescence staining (I) COX IV

immunofluorescence staining, (J) MT-ND4 immunofluorescence staining, (K-V) quantitative statistical analysis of TXNRD1, PI3K, P-PI3K, AKT, P-AKT, P-mTOR, ATP5A, MT-ND4, COX 4, TOMM20, LC3B, ACAN, COL2, MMP13, and ADAMTS5 protein levels. (n = 4 per group, *P < 0.05, **P < 0.01, ***P < 0.001, and ****P < 0.001 by one-way ANOVA with Tukey's multiple comparison test.)

Comment 6. Several aspects of Figure 7 require clarification:

a. The S.O. staining of cartilage in the DMM group appears to be more intense, whereas the staining in the HSPHR group is unexpectedly lighter. The authors should explain the reasons for these observations.

Response: We sincerely appreciate your valuable feedback and deeply regret the unexpected staining outcomes observed in our study. We fully recognize that the hallmark pathological features of DMM involve degeneration of the cartilage matrix and morphological alterations in cartilage structure. In the DMM group, pronounced changes in joint architecture were evident, including the complete loss of joint space. This observation may be closely related to the thickness and uniformity of the tissue sections. Structural disruptions in the joints likely contributed to slight variations in section thickness across certain specimens, which may have led to localized dye over-absorption. For this, we offer our sincere apologies. In response to this concern, we have repeated the experiment using an expanded sample size of eight animals, and the findings were consistent with those presented in the original manuscript. Furthermore, we have systematically reviewed all comparable instances in the manuscript and made appropriate revisions accordingly (Figure R4B-D).

Figure R4. Evaluation of the Efficacy of HSPHR Microspheres in Osteoarthritis Management. (A) Schematic illustration depicting the intra-articular injection of hydrogel microspheres into the knee joint cavity. (B-E) Images of S.O, toluidine blue, hematoxylin, and eosin-stained sections of rat knee joint sections from different treatment subgroups and their quantitative statistical analysis. (F-K) Representative images of COL2, MMP13, TXNRD1, GPX1

immunohistochemical staining of knee joint sections from different treatment subgroups of rats and their quantitative statistical analysis (L-N) Representative images of p-mTOR, p-AKT immunofluorescence staining of knee joint sections from different treatment subgroups of rats and their quantitative statistical analysis. (n = 8 rats per group, * $P < 0.05$, ** $P < 0.01$, *** $P < 0.001$, and **** $P < 0.001$ by one-way ANOVA with Tukey's multiple comparison test.)

b. The immunohistochemical staining results for TXNRD1 and GPX1 in the DMM group are inconsistent with those of other groups, with marked differences in nuclear staining intensity. This raises concerns about the experimental procedures and comparability among groups.

Response: We sincerely appreciate your insightful feedback. As previously mentioned, we used the same specimen, and it is possible that variations in section thickness within this batch led to uneven dye absorption in certain regions. However, it is crucial to emphasize that immunohistochemical staining relies on antigen-antibody binding reactions, which are not affected by excessive dye absorption and should not compromise the consistency of the results. Nevertheless, we acknowledge the experimental inconsistencies and sincerely apologize for them. To address this issue, we have conducted additional experiments, increasing the sample size to eight animals per group. Moreover, we have carefully reviewed all findings that might have caused confusion in the manuscript and have made appropriate revisions, as shown in (Figure R4I–J).

c. Previous studies (Bone Res. 2019 Aug 5;7:23. doi: 10.1038/s41413-019-0062-y) have demonstrated that AKT signaling is activated during OA. However, in this study, the level of p-AKT in DMM mice is significantly decreased. The authors should provide a rationale for this discrepancy.

Response: Thank you for your insightful comments. The precise role of AKT signaling in maintaining articular cartilage homeostasis and its involvement in the progression of osteoarthritis remain largely unclear. Numerous in vitro studies have reported conflicting findings regarding the function of AKT signaling in cartilage homeostasis. While persistent activation of AKT signaling may compromise articular cartilage integrity, other studies

suggest that AKT activation exerts protective effects on chondrocytes (*Nat Commun* 2023;14(1):3159; *Sci Adv* 2024;10(10):eadk6084). Moreover, activation of this pathway has been shown to increase intracellular ATP levels and promote chondrogenic differentiation (*Sci Adv* 2024;10(42):eadp7872). The PI3K/AKT signaling pathway has been demonstrated to exert chondroprotective effects by regulating chondrocyte survival, proliferation, and extracellular matrix synthesis. Furthermore, downstream activation of mTOR has been shown to protect chondrocytes by promoting CoQ10 production through the mTORC1-HMGCR signaling pathway (*Nat Aging* 2025;5(7):1295-1316). This study demonstrates that HSPHR treatment activates the mTOR signaling pathway, thereby modulating cellular energy metabolism. Specifically, it shifts cellular energy production from anaerobic glycolysis to oxidative phosphorylation (OXPHOS), enhances mitochondrial antioxidant capacity, and thereby provides protection to chondrocytes. This protective effect is likely attributed to the moderate and transient activation of phosphorylated AKT (P-AKT).

d. The number of biological replicates per group (n=4) is insufficient to provide robust evidence that HSPHR alleviates OA progression.

Response: We sincerely appreciate your valuable suggestions. In response to your concern regarding the previously inadequate sample size, we have implemented specific measures to address this issue. Specifically, we have conducted additional in vivo experiments, thereby increasing the total sample size per group to eight. All relevant in vivo experiments have been repeated, and thorough quantitative statistical analyses have been carried out. The results consistently demonstrate that HSPHR treatment significantly enhances the expression of selenium-related proteins, namely TXNRD1 and GPX1, in articular cartilage. Moreover, it activates the PI3K-AKT-mTOR signaling pathway in chondrocytes and effectively delays the progression of osteoarthritis in DMM rats, as shown in Figure R4.

Comment 7. The naming of the control groups in Figure 6D is inconsistent between the representative images and the corresponding quantitative data.

Response: Thank you for your valuable suggestions. We have carefully reviewed and corrected the inconsistencies in group names throughout the manuscript.

Comment 8. On page 21, line 8, the authors mention that "HSPHR hydrogel microspheres improve cartilage metabolic...". However, the manuscript does not provide sufficient evidence to clarify whether TXNRD1 restores mitochondrial function via the PI3K-AKT-mTOR pathway, and the causality between these factors is ambiguous. To strengthen the mechanistic interpretation, the authors should consider assessing mitochondrial function and/or the expression of relevant proteins following activation or inhibition of the PI3K-AKT-mTOR pathway. This would provide more direct evidence for the involvement of this signaling axis in the observed effects.

Response : We sincerely appreciate your valuable feedback. In response to your concern regarding the insufficient direct evidence for the regulation of the PI3K-AKT-mTOR signaling pathway, we have carefully reconsidered this issue and conducted additional comprehensive experiments. By utilizing LY294002, a specific inhibitor of PI3K phosphorylation, we further investigated the regulatory effect of HSPHR on the PI3K-AKT-mTOR signaling pathway, as illustrated in Figure R5. The administration of LY294002 significantly attenuated the activation of the PI3K-AKT-mTOR pathway induced by HSPHR (Figure R5A). Further analysis of mitochondrial intrinsic proteins demonstrated that HSPHR effectively restored their expression levels in inflammatory chondrocytes; however, this effect was inhibited by LY294002 treatment (Figure R5B, F–I). Transmission electron microscopy revealed that HSPHR administration promoted the formation of mitochondrial autophagosomes and facilitated the restoration of normal mitochondrial morphology (Figure R5D). Additionally, mitochondrial membrane potential measurements indicated that HSPHR intervention effectively counteracted the IL-1 β -induced disruption of mitochondrial membrane potential (Figure R5E). Moreover, Seahorse extracellular flux analysis was employed to assess the metabolic status of inflammatory chondrocytes following HSPHR treatment. Glycolytic stress testing was performed by measuring the real-time extracellular acidification rate (ECAR) after sequential injections of glucose and 2-deoxy-D-glucose (2-DG), thereby enabling the evaluation of chondrocyte glycolytic parameters. Compared to the Ctrl group, IL-1 β -induced inflammatory chondrocytes exhibited significantly elevated glycolysis and glycolytic reserve. However, in the HSPHR group, these

glycolytic parameters showed a progressive decline. Following treatment with the PI3K inhibitor LY294002, glycolytic reserve was significantly increased, which was accompanied by accelerated mitochondrial damage (Figure R5J). In parallel, real-time oxygen consumption rate (OCR) was measured after sequential administration of oligomycin, carbonyl cyanide-4-(trifluoromethoxy) phenylhydrazone (FCCP), and rotenone, allowing for the evaluation of chondrocyte oxidative phosphorylation (OXPHOS) parameters based on mitochondrial stress assays (Figure R5K). Basal respiration, ATP production, and maximal respiration were notably reduced in the IL-1 β group compared to the control group, whereas OXPHOS parameters gradually increased in the HSPHR group. Notably, LY294002 treatment led to a significant decrease in cellular OXPHOS capacity. These results suggest that HSPHR treatment promotes metabolic reprogramming in inflammatory chondrocytes, shifting energy metabolism from glycolysis toward oxidative phosphorylation, potentially mediated through activation of the PI3K-AKT-mTOR signaling pathway. Subsequently, cells from each group were collected to assess antioxidant capacity. HSPHR treatment enhanced the clearance of excessive reactive oxygen species, including O₂⁻, •OH, and other free radicals (Figure R5L–O). Collectively, these findings indicate that HSPHR therapy restores mitochondrial protein synthesis pathways and reestablishes mitochondrial oxidative phosphorylation, thereby exerting intracellular antioxidant effects and alleviating cartilage damage.

Figure R5. HSPHR activates the PI3K-AKT-mTOR pathway to restore mitochondrial oxidative phosphorylation and mitigate chondrocyte damage. (A) WB bands of PI3K, P-PI3K, AKT, P-AKT, and P-mTOR WB bands. (B) WB bands of ATP5A, MT-ND4, COX 4, TOMM20, and LC3B WB. (C) WB bands of ACAN, COL2, MMP13, and ADAMTS5 bands. (D) TEM of mitochondria. (E) JC-1 staining. (F) TOMM20 immunofluorescence staining. (G) ATP5A immunofluorescence staining. (H) COX IV immunofluorescence staining. (I) MT-ND4 immunofluorescence staining. (J) Real-time ECARs of chondrocytes during glycolytic stress testing, (K) Real-time OCR in chondrocytes during mitochondrial stress testing. (L–O) Scavenging efficiency of chondrocytes against three typical free radicals PTIO,

ABTS, •OH, and DPPH.(n = 4 independent experiments per group, *P < 0.05, **P < 0.01, ***P < 0.001, and ****P < 0.001 by one-way ANOVA with Tukey's multiple comparison test.)

Furthermore, when the experiment was repeated using 740Y-P (a phospho-activated PI3K activator), we observed that the therapeutic effect of HSPHR was comparable to that of direct activation by 740Y-P. This treatment effectively restored mitochondrial intrinsic protein expression levels and normalized mitochondrial morphology and function (Figure R6). In summary, our findings clearly indicate that the mechanism of action of HSPHR is closely linked to the activation of the PI3K-AKT-mTOR signaling pathway.

Figure R6. HSPHR activates the PI3K-AKT-mTOR pathway to restore mitochondrial oxidative phosphorylation and mitigate chondrocyte damage. (A) WB bands of PI3K, P-PI3K, AKT, P-AKT, and P-mTOR. (B) WB bands of ATP5A, MT-ND4, COX 4, TOMM20, and LC3B. (C) WB bands of ACAN, COL2, MMP13, and ADAMTS5. (D–P) Quantitative analysis of the aforementioned WB bands. (Q) TEM of mitochondria. (R–S) JC1 staining and its statistical analysis. (T) Real-time OCR during mitochondrial stress testing and

semi-quantitative analysis of basal respiration, maximal respiration, and ATP production in chondrocytes. (n = 4 per group, *: $*P < 0.05$, **: $**P < 0.01$, ***: $***P < 0.001$, and ****: $****P < 0.001$ by one-way ANOVA with Tukey's multiple comparison test.)

Reviewer #2 (Remarks to the Author):

The manuscript presents a novel synthesis of a ROS/MMP13 dual-responsive organic double selenium hydrogel microsphere to realize early OA local microenvironmental response by utilizing the pathological characteristics of osteoarthritis, and at the same time, the organic selenium can effectively supplement the selenoprotein level in the cartilage, synovium and subchondral bone. Intra-articular injection of HSPHR not only reduced cartilage degeneration, improved synovial hyperplasia, and delayed sclerosis of the subchondral bone in the post-traumatic OA model, but also accelerated the formation of new cartilage in the cartilage defect model. While the study is innovative and holds significant potential, I would like to provide the following constructive suggestions to enhance the scientific rigor and presentation clarity of this manuscript before it can be considered for publication.

Response: We are grateful to the Reviewer for the constructive Comment, which have improved our manuscript (details below).

***Comment 1.** The "Introduction" section is too long. Authors should focus only on the most important aspects for a good understanding of the developed work.*

Response: Thank you for your valuable suggestions. We have carefully revised the introduction section of the manuscript to enhance its clarity and focus on the key aspects of the research. This section has been marked in the revised manuscript.

***Comment 2.** The authors propose that their "three birds with one stone" modified organic diselenide hydrogel microspheres can simultaneously treat multiple aspects of osteoarthritis. However, this conceptual framework requires clarification for several reasons:*

We recommend either:

- a) Providing direct experimental evidence for triple-targeting efficacy, or*
- b) Modifying the conceptual framework to better reflect the actual therapeutic mechanisms observed.*

This would significantly strengthen the manuscript's mechanistic claims while maintaining scientific rigor. The current presentation risks overinterpretation of the technology's targeting capabilities.

Response:

a) Your suggestions are highly insightful. HSPHR can only exert therapeutic effects in response to elevated levels of ROS and MMP13 within the joint. Substantial ROS production within tissues may lead to both macrophage inflammation and excessive osteoclast activation. Therefore, HSPHR demonstrates synergistic therapeutic effects across multiple tissues, which is fully consistent with our original objective. Our initial aim was to develop a therapeutic strategy targeting three distinct intra-articular tissues. By leveraging its unique ROS-responsive properties, HSPHR enhances intracellular selenoprotein expression, thereby effectively modulating all three distinct pathological tissues associated with OA.

b) Our research primarily focused on the therapeutic effects of selenoproteins on chondrocytes, with limited exploration of synovial macrophages and osteoclasts. Given the premise that HSPHR exerts therapeutic effects on both synovial macrophages and osteoclasts, we further investigated the potential mechanisms underlying HSPHR treatment in these two cell types. This study aimed to determine whether HSPHR alleviates synovial inflammation, suppresses osteoclast activation, and restores chondrocyte activity through a shared regulatory mechanism. Notably, transcriptomic sequencing revealed that HSPHR's therapeutic effects simultaneously activate the PI3K-AKT-mTOR signaling pathway in chondrocytes, synovial macrophages, and subchondral osteoclasts (Figure R7B, R9B). To validate this hypothesis, we conducted follow-up experiments. By selectively activating and inhibiting the PI3K-AKT-mTOR pathway in osteoclasts and synovial macrophages, respectively, we found that HSPHR therapy activates this pathway to promote M2 polarization in synovial macrophages, and that pathway inhibition counteracts the therapeutic effects of HSPHR (Figure R7D-Q, R8A-N). Furthermore, activation of the PI3K-AKT-mTOR pathway effectively mitigates early excessive osteoclast activity in subchondral bone (Figure R9D-Q). These findings are consistent with our research hypothesis: HSPHR intervention exerts therapeutic effects on three cell types simultaneously, akin to achieving a "triple-target" outcome with a single intervention.

Figure R7. HSPHR regulates synovial macrophage polarization via the PI3K-AKT-mTOR signaling pathway. (A) Volcano plot of differentially expressed genes between HSPHR and CTRL. (B) KEGG pathways enriched by differentially expressed genes between HSPHR and CTRL. (C) ESGA diagram of KEGG pathways enriched. (D–I) CD68/CD86, CD68/CD206 immunofluorescence, representative flow cytometry channels, and quantitative analysis. (J–Q)

Representative Western blot images and quantitative analysis of CD206, CD86, ARG1, NOS2, P-PI3K, P-AKT, and P-mTOR after HSPHR treatment of synovial macrophages. (n = 4 per group, *: **P* < 0.05, ***P* < 0.01, ****P* < 0.001, and *****P* < 0.001 by one-way ANOVA with Tukey's multiple comparison test.)

Figure R8. The PI3K-AKT-mTOR pathway activation induces M2 macrophage polarization. (A, D-N) Western blots and analysis show CD206, CD86, ARG1, NOS2, P-PI3K, P-AKT, and P-mTOR levels in synovial macrophages post-HSPHR treatment. (B-C) Immunofluorescence and flow cytometry display CD68/CD86, CD68/CD206 markers. (n = 4 per group, *: **P* < 0.05, ***P* < 0.01, ****P* < 0.001, and *****P* < 0.001 by one-way ANOVA with Tukey's multiple comparison test.)

Figure R9. HSPHR regulates osteoclast activation through the PI3K-AKT-mTOR signaling pathway. (A) Volcano plot of differentially expressed genes between HSPHR and CTRL. (B) KEGG pathways enriched by differentially expressed genes between HSPHR and CTRL. (C) ESGA plot of KEGG pathways enriched. (D–F) Representative images and quantitative analysis of Trap staining and β -actin staining after HSPHR treatment of osteoclasts. (E, I–L) Representative WB images and quantitative analysis of CTSK, CD40L, P-PI3K, P-AKT, and

P-mTOR following HSPHR treatment of osteoclasts, along with quantitative statistical analysis. (H, M) Trap staining and representative Western blot images of CD206, CD86, ARG1, NOS2, P-PI3K, P-AKT, and P-mTOR following HSPHR and 740Y-P treatment of osteoclasts. (n = 4 per group, *: * $P < 0.05$, ** $P < 0.01$, *** $P < 0.001$, and **** $P < 0.001$ by one-way ANOVA with Tukey's multiple comparison test.)

Comment 3. While the abstract states that organic selenium supplementation restores selenoprotein levels across multiple joint tissues (cartilage, synovium, and subchondral bone), the study primarily demonstrates mechanistic effects in cartilage without providing selenium quantification data for synovium or subchondral bone. This limitation should be acknowledged, as it affects the interpretation of the compound's proposed multi-tissue targeting capability.

Response: We sincerely appreciate your valuable suggestions. We sincerely apologize for the issue concerning the manuscript. To further support our findings, we have added detailed experimental data in the relevant section, with the aim of elucidating and refining the therapeutic mechanisms mediated by HSPHR in cartilage, synovium, and subchondral bone. We separately measured selenoprotein levels in cartilage and synovial tissues after DMM 8 weeks. Results demonstrated that HSPHR treatment effectively restored GPX1 and TXNRD1 protein expression in both cartilage and synovial tissue (Figure R10A-B, E-F). Identical outcomes were observed in subchondral bone tissue at 2 weeks after DMM (Figure R10C-D). These findings indicate that HSPHR can upregulate selenoproteins across different tissues. Subsequently, we analyzed the underlying mechanisms in detail (Figures R6-9). These regulatory effects all point to a common regulatory pathway involving PI3K-AKT-mTOR.

Figure R10. HSPHR Treatment Effects on Selenoprotein Levels in Cartilage, Synovium, and Subchondral Bone Following DMM. (A-B) Representative histochemical staining and quantitative analysis of GPX1 and TXNRD1 in cartilage tissue at 8 weeks after DMM. (C-D) Representative immunohistochemical staining and quantitative analysis of GPX1 and TXNRD1 in subchondral bone at 2 weeks after DMM surgery. (E-F) Representative immunohistochemical staining and quantitative analysis of GPX1 and TXNRD1 in synovium at 8 weeks after DMM surgery. (n = 3-8 per group, *: $P < 0.05$, ** $P < 0.01$, *** $P < 0.001$, and **** $P < 0.001$ by one-way ANOVA with Tukey's multiple comparison test.)

Comment 4. While the authors showed HSPHR's inhibitory effect on osteoclast formation in vitro in Supplementary Figure 6, the in vivo study only evaluated late-stage OA pathology (8

weeks post-operation). Since osteoclasts are known to mediate early-stage OA pathogenesis, could the authors address why osteoclast activity was not monitored at earlier time points? This information would strengthen the mechanistic connection between HSPHR's cellular effects and its therapeutic outcomes.

Response: We sincerely appreciate your valuable suggestions and fully align with your perspective. In the early stages of osteoarthritis, subchondral bone may exhibit hyperactive osteoclast function. To further investigate this phenomenon, we conducted a new series of animal experiments using Sham, DMM, HAMA, and HSPHR groups. Subchondral bone measurements and bone density analyses were performed on rats two weeks after DMM surgery. The results demonstrated that rats in the DMM group exhibited excessive osteoclast activation at two weeks post-surgery, whereas HSPHR treatment significantly attenuated this activation and restored the initially reduced bone density (Figure R11A, C–E). Further immunohistochemical staining of knee joint tissues from two-week-old rats revealed markedly elevated MMP9 expression in the DMM group at this time point. Following HSPHR treatment, osteoclast activation was significantly reduced, and MMP9 expression returned to baseline levels (Figure R11B, F). These findings suggest that HSPHR therapy effectively modulates osteoclast activation in subchondral bone during the early stages of arthritis. The underlying mechanism may involve the scavenging of ROS produced by cartilage tissue, thereby limiting osteoclast activation in subchondral bone. Alternatively, HSPHR may exert direct effects on osteoclasts. Evidence from previous studies indicates that increased selenoprotein synthesis can directly regulate osteoclast activity and reverse the process of osteoclast activation (*Adv Mater* 2024;36(27):e2401620).

Figure R11. HSPHR suppresses osteoclast activation in the early stage of osteoarthritis. Two weeks after DMM surgery, representative images and quantitative analysis of (A) subchondral bone remodeling, (B, F) MMP9 immunohistochemical staining in the Sham, DMM, HAMA, and HSPHR groups, and (C-F) subchondral bone trabecular separation, bone volume fraction, and trabecular density. (n = 8 per group, *: $P < 0.05$, **: $P < 0.01$, ***: $P < 0.001$, and ****: $P < 0.001$ by one-way ANOVA with Tukey's multiple comparison test.)

Comment 5. The Graphic Abstract does not clearly convey the "three-in-one" concept, nor does it show the microspheres' impact on other cells.

Response: Thank you very much for your insightful suggestions. Our research mainly investigates the degenerative process of osteoarthritis. Notably, our experimental results revealed that HSPHR treatment exerts beneficial regulatory effects on both osteoclasts and synovial macrophages. We sincerely apologize for the insufficient presentation of these findings in the original figure abstract. Accordingly, we have revised the figure abstract to more clearly depict the mechanism through which HSPHR regulates selenoprotein expression to restore articular joint homeostasis.

Comment 6. To further explore the advantageous effects of HSPHR on cartilage regeneration, the authors developed an in vivo model featuring a full-thickness cartilage defect. Chondrogenic progenitor cells were incorporated into microspheres of HSPHR in order to enhance the therapeutic outcomes for cartilage repair. The study primarily elucidates the role of selenoproteins (particularly GPX1 and TXNRD1) in pathological cartilage, as demonstrated in both the main text and schematic diagrams. However, the relationship between selenoproteins and chondrocyte differentiation remains unclear. Several critical questions need to be addressed:

a) Molecular Mechanisms: Are GPX1 and TXNRD1 similarly involved in chondrocyte

differentiation and cartilage homeostasis? What specific selenoproteins regulate chondrogenic differentiation pathways?

b) Experimental Evidence: The study would benefit from in vitro experiments using:

1) Chondrocyte differentiation models;

2) Selenoprotein-specific knockdown/overexpression systems;

3) Comprehensive analysis of redox regulation during differentiation.

Response: a) We sincerely appreciate your valuable insights, which have enabled us to further explore the mechanisms through which selenoproteins promote chondrogenic differentiation. Accumulating evidence suggests that the selenoprotein family plays a crucial role in early cartilage development, involving the coordinated actions of multiple selenoproteins. Recent studies indicate that GPX1 actively contributes to chondrogenic differentiation, primarily by alleviating oxidative stress in chondrogenic precursor cells, enhancing cartilage matrix synthesis, and facilitating the differentiation process (*Mater Today Bio* 2023;23:100864; *Arthritis Rheumatol* 2014;66(12):3349-58). As a homologous member of the selenoprotein family, TXNRD1 also participates in cellular redox metabolism across various tissues (*Nat Aging* 2024;4(2):185-197). Our findings demonstrate that TXNRD1 exerts a more pronounced protective effect on cartilage and exhibits enhanced chondrogenic differentiation capacity (Figure R12). This effect may be associated with the activation of the PI3K-AKT-mTOR signaling pathway downstream of TXNRD1, as discussed in our manuscript. Activation of this pathway restores mitochondrial function impaired by inflammatory injury, shifting cellular metabolism from glycolysis to oxidative phosphorylation, thereby more effectively promoting chondrogenic differentiation.

b) Thank you for your advice. To investigate whether GPX1 and TXNRD1 can promote chondrogenesis, we designed experiments involving the transfection of overexpression plasmids of GPX1 and TXNRD1 into ATDC5 chondrogenic precursor cells. Following the induction of chondrocyte pellet formation, Alcian blue and eosin staining were performed. The results demonstrate that overexpression of both GPX1 and TXNRD1 enhances chondrogenic differentiation under inflammatory conditions, with TXNRD1 overexpression showing a more pronounced promotion of cartilage matrix secretion (Figure R12A-D). To

further validate these findings, we conducted siRNA-mediated knockdown of GPX1 and TXNRD1 in ATDC5 cells, followed by induction of chondrogenic differentiation. Both siGPX1 and siTXNRD1 led to a reduction in cartilage matrix production, with TXNRD1 knockdown resulting in a significantly greater decrease, thereby severely impairing chondrogenic differentiation (Figure R12E-H). Collectively, these findings indicate that both GPX1 and TXNRD1 contribute to the promotion of chondrogenic differentiation under osteoarthritic conditions. Notably, our study reveals that TXNRD1 exerts a more pronounced protective effect, representing the first observation of its kind during chondrogenic differentiation. Subsequently, we analyzed changes in ATDC5 cells' antioxidant capacity during chondrogenic differentiation, assessing scavenging abilities following TXNRD1 overexpression and HSPHR treatment. Results indicate that both HSPHR intervention and TXNRD1 overexpression enhance cellular antioxidant capacity, improving chondrocyte scavenging efficiency against three representative free radicals PTIO, ABTS, \bullet OH, and DPPH (Figure R12I-L).

Figure R12. GPX1 and TXNRD1 restore cellular antioxidant capacity and promote in vitro

chondrogenic differentiation. (A-D) Representative images and quantitative statistics of S.O. and A.B. staining in GPX1 and TXNRD1-overexpressing chondrocyte chips under IL-1 β (E-H) IL-1 β conditions. (E-H) Representative images and quantitative analysis of S.O. and A.B. staining in cartilage chips with GPX1 and TXNRD1 knockdown under IL-1 β treatment. (I-L) Radical scavenging efficiency of cartilage chips after TXNRD1 overexpression against three typical free radicals and DPPH. (n = 3 per group, *: * P < 0.05, ** P < 0.01, *** P < 0.001, and **** P < 0.001 by one-way ANOVA with Tukey's multiple comparison test.)

Comment 7. Figure 4, the authors report solid-state NMR characterization of HA-Se-Se, but the presented spectrum shows only a single peak with no observable selenium signal. This raises several technical and interpretational concerns that need to be addressed.

Response: We sincerely apologize for any confusion concerning our results. In Figure 4, solid-state NMR was employed to analyze the selenium spectrum of HSPHR. The sample preparation process, which involved repeated grinding and impurity removal, may have led to partial oxidation of the test samples. Moreover, due to the inherent limitations of solid-state NMR, broad spectral peaks were obtained, thereby reducing measurement precision. To address this, we supplemented our analysis with liquid NMR and Raman spectroscopy to more accurately characterize the specific Se-Se bonds in HSPHR. The ^{77}Se NMR spectrum revealed a distinct Se-Se peak at 290.23 ppm (Figure R13A). Likewise, Raman spectroscopy identified a characteristic Se-Se metallic-like vibrational peak at 254 cm^{-1} (Figure R12B). Further FTIR analysis also detected characteristic Se-Se absorption bands (Figure R13C), and XPS analysis confirmed the presence of Se 3d signals (Figure R13D–E). Collectively, these findings provide strong evidence that the synthesized HSPHR microspheres contain abundant Se-Se bonds.

Figure R13. Characterization of HSPHR. (A) ^{77}Se NMR spectrum of HSPHR, (B) Raman spectrum comparison between HSPHR and HAMA, (C) FTIR spectrum comparison between HSPHR and HAMA, (D–E) XPS spectrum comparison between HSPHR and HAMA.

Comment 8. While the manuscript investigates selenium-based therapy for OA, several critical concerns regarding its novelty and mechanistic depth need to be addressed:

Innovation Gap:

The therapeutic use of selenium nanoparticles in OA has been extensively reported (e.g., studies by DOI: 10.1016/j.mtbio.2023.100864; DOI: 10.1016/j.biomaterials.2025.123195; DOI: 10.1016/j.jcis.2025.137612).

The current work must better differentiate itself by:

Response: Thank you for your insightful comments. Current selenium-based studies on articular cartilage repair mainly emphasize ROS scavenging and the modulation of the local microenvironment. The OHA/HA-ADH@SeNPs system reported in the *Materials Today Bio* presents notable innovations, including the sustained release of SeNPs and the targeted activation of GPX1. Nevertheless, the underlying mechanisms of selenoprotein-based therapies in OA remain insufficiently explored. The Se/PRP-OGel system introduced in the *Journal of Colloid and Interface Science* employs zero-valent selenium nanoenzymes to eliminate ROS and utilizes PRP-derived growth factors to promote cartilage regeneration. However, it does not clarify the specific selenoprotein pathways involved in regulating mitochondrial quality control in OA chondrocytes, nor does it provide a systematic evaluation of the synergistic pathological interactions between synovial macrophages and subchondral

osteoclasts. In the *Biomaterials* study, HAsenNs@AHAMAHMs were developed as "cascade-targeted" microspheres, highlighting CD44-mediated uptake and enhanced mitochondrial oxidative phosphorylation. Yet, similar to previous studies, this work primarily focuses on SeNPs release and antioxidant effects, without offering direct evidence regarding which selenoproteins are restored or how their upstream signaling pathways contribute to mitochondrial homeostasis. Moreover, it lacks a comprehensive intervention strategy targeting the triad of pathological units—synovium, cartilage, and subchondral bone.

This study proposes and validates an innovative modification of organic selenium (Se-Se) dual-responsive microspheres, HSPHR. The modification achieves three key innovations: (1) dual ROS/MMP13 responsiveness that aligns with osteoarthritis (OA) pathology, enabling on-demand drug release; (2) enhanced selenium absorption efficiency and improved biosafety. It is demonstrated that TXNRD1 activates the PI3K-AKT-mTOR signaling pathway, shifting cellular metabolism from glycolysis to OXPHOS. This metabolic reprogramming alters cellular energy metabolism and facilitates the recovery of impaired mitochondrial function. (3) The regulatory role of the PI3K-AKT-mTOR pathway is shown to operate across three cellular dimensions, simultaneously alleviating cartilage degeneration, suppressing synovial macrophage inflammation, and inhibiting early osteoclast activation. This approach represents a novel therapeutic strategy for OA that has not been previously reported in the literature.

Selenoprotein Analysis Limitations:

The cartilage selenoprotein expression profiling GPX1/TXNRD1 largely replicates prior work without advancing the field.

Response: Thank you for your valuable feedback. It is well established that GPX1 and TXNRD1 have been extensively studied and are known to exert significant chondroprotective effects, primarily by enhancing cellular antioxidant capacity. In this study, we show that HSPHR microspheres not only effectively upregulate the expression of GPX1 and TXNRD1, but also demonstrate that TXNRD1 restores mitochondrial oxidative phosphorylation through the activation of the PI3K-AKT-mTOR signaling pathway, thereby reversing the altered energy metabolism in OA chondrocytes. Moreover, we further confirm that the PI3K-AKT-mTOR signaling pathway plays a regulatory role in two additional cell types

associated with osteoarthritis—synovial macrophages and osteoclasts.

Mechanistic Shortcomings:

Which specific selenoproteins mediate the therapeutic effects in cartilage;

How their regulation differs across joint tissues (synovium vs. cartilage vs. bone);

The downstream pathways linking selenoproteins to chondroprotection.

Response: We sincerely appreciate your valuable suggestions. Our research primarily focused on the therapeutic effects of selenoproteins on chondrocytes, with a limited investigation into their impact on synovial macrophages and osteoclasts. As previously outlined, by examining the broad therapeutic potential of selenoproteins across multiple tissue types, combined with our transcriptomic sequencing data, we further explored the effects of HSPHR on synovial macrophages and osteoclasts. Notably, transcriptomic analysis revealed that HSPHR activates the PI3K-AKT-mTOR signaling pathway simultaneously in chondrocytes, synovial cells, and subchondral osteoclasts (Figures R7B, R9B). To validate this finding, we conducted follow-up experiments. By selectively activating and inhibiting the PI3K-AKT-mTOR pathway in osteoclasts and synovial macrophages, respectively, our results indicate that HSPHR therapy promotes M2 polarization of synovial macrophages through activation of this pathway, and that pathway inhibition attenuates the therapeutic efficacy of HSPHR (Figures R7D-Q, R8A-N). Concurrent activation of the PI3K-AKT-mTOR pathway effectively suppresses early excessive osteoclast activity in subchondral bone (Figure R9D-Q), thereby supporting our hypothesis that HSPHR exerts therapeutic effects on three distinct cell types, akin to a multi-target intervention strategy. Our study centers on the selenoproteins GPX1 and TXNRD1 to identify a novel therapeutic mechanism with potential for comprehensive joint tissue protection.

Notably, the PI3K-AKT-mTOR signaling pathway demonstrates a range of functional roles across different diseases and cell lines. Our novel strategy is centered on the development of an organic selenium-containing hydrogel microsphere capable of simultaneously activating the PI3K-AKT-mTOR pathway in three distinct cell types. This multifunctional approach exerts a comprehensive therapeutic effect by promoting chondrocyte regeneration, modulating macrophage-mediated immunity, and suppressing

osteoclast activation. By integrating these mechanisms, our approach effectively targets multiple pathological processes in osteoarthritis, representing a meaningful step forward in the advancement of clinical therapies.

Reviewer #3 (Remarks to the Author):

Comment:

This study developed ROS/MMP13-responsive selenium microspheres to target OA pathology. HSPHR restores selenoproteins and activates PI3K-AKT-mTOR-mediated mitophagy, mitigating cartilage degeneration, synovitis, and subchondral sclerosis in post-traumatic OA models while enhancing regeneration in cartilage defects. While the work addresses an important clinical issue, it falls short in several critical areas, particularly regarding novelty and experimental depth. As such, the current version of the manuscript is not suitable for publication in Nature Communications.

Response: We sincerely appreciate the reviewer's constructive comments, which have contributed to the improvement of our manuscript (see details below).

Comment 1. The abstract and the introduction should be rewritten, the novelty of the work is not highlighted. What does "Three birds with one stone" mean in the title? It needs to be clearly stated in the introduction.

Response: We sincerely appreciate your valuable feedback concerning the identified shortcomings. In response, we have thoroughly revised the abstract and introduction to more clearly emphasize the role of selenium-containing hydrogels in activating the PI3K-AKT-mTOR signaling pathway. This activation enables a synergistic therapeutic strategy that addresses three key pathological aspects of osteoarthritis through a triple-action mechanism.

Comment 2. The graphical abstract fails to effectively convey the "Three birds with one stone" concept and requires redesign to explicitly highlight the innovative therapeutic strategy of the HSPHR system.

Response: We sincerely appreciate your valuable suggestions. Our research primarily focuses on the degenerative processes affecting articular cartilage. Notably, our experimental findings demonstrate that HSPHR treatment exerts beneficial regulatory effects on both osteoclasts and synovial macrophages. We apologize for the inadequate illustration of these results in the original figure abstract. Accordingly, we have revised the figure abstract to more clearly and

accurately depict the mechanism by which HSPHR modulates selenoprotein expression to restore homeostasis in articular cartilage.

Comment 3. The article uses a great many abbreviations, some of which are not explained at all (e.g. CEMIP, HSR, DMM, Ser, PTEN, WB, etc.). The abbreviations of all text are very confusing. Full names should be given when an abbreviation first appears in the text. Additionally, the manuscript contains redundant abbreviation definitions (e.g., osteoarthritis (OA) appears with parenthetical clarification multiple times) and inconsistent terminology for key abbreviations. Specifically, the term HSPHR is ambiguously defined as both: “dual-responsive organicdouble selenium hydrogel microsphere (HSPHR)”

“dual-responsive selenium nanoparticle hydrogel microspheres (HSPHR) ”

Response: We sincerely apologize for the use of ambiguous terminology and the inclusion of inadequately defined abbreviations. We have carefully revised the manuscript to improve clarity and to better support its evaluation by future researchers.

Comment 4. Given that most prior studies refer to “SeNPs” as inorganic selenium nanoparticles, while this work focuses on organic selenium, what specific form does “SeNPs” denote in this context? Please discuss the advantages of organic selenium compared to inorganic selenium.

Response: Selenium is an essential trace element for the human body. Inorganic selenium typically lacks C-Se covalent bonds and is commonly found in the form of sodium selenite (Na_2SeO_3). In this study, selenocysteine (Se-Cys), which contains C-Se covalent bonds, was synthesized. Following cellular uptake, Se-Cys can be directly incorporated into selenium-containing proteins in the body, participating in processes such as biosynthesis and redox balance. Organic selenium has a wider safety margin and is generally regarded as safer than its inorganic counterpart. Research investigating the effects of different selenium compounds on human cells has shown that inorganic selenium is more likely to induce cell growth inhibition and cell death compared to organic selenium (*Bioinorg Chem Appl* 2014;2014:923834). In contrast to inorganic selenium, which exhibits low bioavailability and a higher risk of carcinogenicity, organically synthesized selenium demonstrates broader potential in the development of bioactive materials. Our findings suggest that organically bound selenium exhibits superior antioxidant activity in vitro and high direct conversion efficiency within cells. It facilitates the more efficient synthesis of antioxidant proteins such as GPX1 and TXNRD1, thereby effectively slowing the progression of osteoarthritis and promoting cartilage regeneration.

Comment 5. The authors claim SeNPs reverse OA cartilage degeneration by restoring antioxidant pathways. However, given the significant influence of nanoparticle morphology and size on bioactivity, the current characterization of these parameters is insufficient and requires enhancement. For instance, comprehensive characterization techniques such as TEM

(Transmission Electron Microscopy) for morphology visualization, DLS (Dynamic Light Scattering) for size distribution analysis, and zeta potential measurement for surface charge evaluation should be employed to provide a more accurate and detailed understanding of the SeNPs' physicochemical properties. Additionally, justification is needed for selecting 20 nM and 50 nM concentrations in validating SeNPs' effects on chondrocyte matrix metabolic homeostasis.

Response: We sincerely appreciate the reviewers' valuable comments. The characterization of biomaterials constitutes a critical component of our research. SeNPs were thoroughly characterized in our previous studies (*J Nanobiotechnology* 2024;22(1):634), and we apologize for any confusion caused by the omission of this information in the current manuscript. In response to the reviewers' suggestions, we have now included a detailed characterization of the SeNPs synthesized in this study. Transmission electron microscopy was employed to analyze the morphological structure of SeNPs (Figure R1A). The particle size of SeNPs was determined to be approximately 250 nm, which is consistent with the results obtained from dynamic light scattering measurements (Figure R1C). To confirm the elemental composition of the nanoparticles, energy-dispersive X-ray spectroscopy (EDX) was conducted, revealing a uniform distribution of selenium on the surface of SeNPs (Figure R1B). As illustrated in Figure R1D, SeNPs exhibit a negative zeta potential of 11.7 mV, indicating favorable surface charge stability.

Regarding the selection of SeNPs concentration, our prior study investigated the biosafety profile of SeNPs across a range of concentrations (*J Nanobiotechnology* 2024;22(1):634). The findings demonstrated that concentrations exceeding 50 ng/mL significantly inhibited cell proliferation and reduced cell viability. However, this concentration also yielded the most favorable therapeutic outcomes. Consequently, the optimal therapeutic concentration of SeNPs employed in this study was established as 50 ng/mL. We apologize for the imprecise unit notations previously used in the figure captions and have revised all ambiguous annotations for clarity.

Comment 6. Please standardize the capitalization of the first letters of the x- and y-axis labels, as well as the font style and size in all figures, and define the abbreviations for each group.

Response: Thank you very much for your rigorous formatting guidelines for the manuscript. We have carefully revised all fonts and formatting elements to better support the subsequent review process by the experts.

Comment 7. The presentation of many data in the manuscript is not substandard. All morphological images must include scale bars, such as Figures 2A,3E-M, 5A-D, 7B-M, S7E-I, and S8. All fluorescently stained images should provide information on the fluorescent probes used, such as Figures 5A-D, 7L-M, S6B, and S7E-F, etc.

Response: Thank you for your valuable feedback. We have relabeled all the images mentioned above and conducted a thorough review of all other images.

Comment 8. Selenium plays a critical role in maintaining redox homeostasis by participating in antioxidant processes and regulating selenoprotein synthesis. However, this study lacks experimental evidence demonstrating how SeNPs modulate redox homeostasis.

Response: We sincerely appreciate your insightful comments. Our study further investigated the mechanisms through which HSPHR enhances cellular antioxidant capacity. Experimental findings demonstrated that the in vitro antioxidant capacity of SeNPs is significantly inferior to that of HSPHR. In alignment with previous studies, we confirmed that HSPHR activates the PI3K-AKT-mTOR signaling pathway, as evidenced by the use of both PI3K phosphorylation inhibitors and activators. Furthermore, we comprehensively analyzed the intracellular antioxidant mechanisms of HSPHR, with particular focus on mitochondrial protein expression, morphological changes, shifts in energy metabolism, and the rate of superoxide anion clearance (Figures R5–R6). Collectively, HSPHR activation promotes the restoration of mitochondrial protein synthesis pathways and the re-establishment of mitochondrial oxidative phosphorylation, thereby exerting potent intracellular antioxidant effects and mitigating cartilage damage.

Comment 9. The western blot results in Figure 1D, and 6B lack reliability due to inconsistent loading controls.

Response: We sincerely apologize for the lack of a blank control in the original study. To

address this limitation, we have redesigned the experiments presented in Figure 6 by introducing a Ctrl group without IL-1 β as a blank control. All experiments—including Western blotting, immunofluorescence, and transmission electron microscopy—have been repeated with this revised setup (Figure R3). The results remain consistent with our previous findings: HSPHR exerts its effects by upregulating the selenoprotein TXNRD1, which subsequently activates the PI3K-AKT-mTOR signaling pathway. This activation leads to the restoration of normal mitochondrial metabolism and, consequently, the normalization of chondrocyte metabolism.

Comment 10. The therapeutic efficacy evaluation of HSPHR microspheres in early-stage osteoarthritis lacks critical validation through imaging modalities (e.g., Micro-CT, MRI) and behavioral assessments (e.g., gait analysis), undermining the clinical relevance of reported histological improvements.

Response: Thank you for your valuable suggestions. As an exercise-related degenerative disease, osteoarthritis requires early behavioral assessment and imaging observation for timely intervention. Considering the challenges associated with conducting behavioral experiments in rats, we utilized 6-week-old mice for DMM modeling and assessed behavioral changes eight weeks after HSPHR knee joint injection therapy. Spontaneous activity levels were recorded over a 3-minute period using the open field test (OFT). Observations showed that, compared to the sham-operated group, DMM-operated mice exhibited significantly reduced relative activity levels, active time, total activity distance, and average velocity. These impairments were markedly reversed following HSPHR treatment (Figure R14A–E). Moreover, we evaluated pain and gait status using footprint analysis. The results indicated that HSPHR treatment significantly alleviated pain in DMM-induced mice, leading to improvements in stride and step length, as well as reductions in fore- and hindlimb print lengths (Figure R14J–K). Additionally, we conducted micro-CT scans of the subchondral bone at 2 weeks (Figure R10A, C–E) and 8 weeks (Figure R14F–H) after DMM surgery. The findings revealed that HSPHR effectively suppressed the abnormal activation of subchondral osteoclasts during the early stages of osteoarthritis, thereby protecting against pathological progression in subchondral bone and preventing subchondral bone sclerosis. In summary,

compared to the DMM group, the HSPHR-treated group demonstrated enhanced spontaneous locomotor activity and significantly improved physical performance.

Figure R14. HSPHR can alleviate subchondral bone sclerosis and enhance physical mobility in OA. (A) Representative trajectory plots displayed by activity intensity thresholds, (B–E) Changes in spontaneous activity, including relative activity level, activity duration, distance traveled, and average speed, (F) Subchondral bone micro-CT reconstruction, (G–H) Subchondral bone bone volume fraction and trabecular thickness, (I) Representative footprint images across groups, (J–K) Changes in relative stride length and stride frequency in HSPHR-treated mice after DMM surgery. (n = 8 per group, *: $P < 0.05$, ** $P < 0.01$, *** $P < 0.001$, and **** $P < 0.001$ by one-way ANOVA with Tukey’s multiple comparison test.)

Comment 11. Figures 7L-M, S2, S5, S6A-B, S7E, and S9 exhibit poor quality requiring enhanced resolution and annotation. Critically, Figure S5 lacks organ labeling in HSPHR biodistribution images.

Response: We sincerely appreciate the suggestions provided. All images previously mentioned have been re-uploaded in high-definition format. We extend our apologies for any inconvenience this may have caused during your review. Furthermore, concerning the potential concern regarding drug retention in the liver, we conducted a comprehensive biosafety evaluation. Serum samples collected from rats at two and four weeks post-DMM administration were analyzed to assess the long-term safety of microsphere injection (Figure R15). The findings demonstrate that HSPHR injection therapy does not induce visceral damage.

Figure R15. *In vivo* biological safety of HSPHR. (A) Blood biochemical parameters in the HSPHR group versus the Sham group 2 weeks after DMM surgery. (B) Blood biochemical parameters in the HSPHR group versus the Sham group 4 weeks after DMM surgery. (n = 6per group, *: * $P < 0.05$, ** $P < 0.01$, *** $P < 0.001$, and **** $P < 0.001$ by one-way ANOVA with Tukey's multiple comparison test.)

Comment 12. The discussion section is superficial and lacks depth. The authors should provide a more thorough analysis of their results, comparing their findings with existing literature and highlighting the implications of their work for clinical applications.

Response: We sincerely appreciate your valuable suggestions. Accordingly, we have expanded the manuscript to incorporate a more comprehensive discussion, highlighting the distinctions between existing research and the present study, as well as exploring the potential clinical applications of this work in the future.

Comment 13. This manuscript needs a revision. The writing of this paper is too rough. The manuscript requires thorough language revision due to frequent formatting errors (e.g., incorrect subscript/superscript notation in chemical formulas like Na₂SeO₃ and H₂O₂), spelling mistakes (such as ofor, and SeNPss, etc.), and grammatical inconsistencies. Comprehensive English editing is recommended to ensure scientific rigor and clarity.

Response: We sincerely apologize for any difficulties you may have encountered while reading the manuscript. We have carefully revised and polished the document to enhance its clarity, thereby facilitating a more effective expert review and enabling the provision of valuable feedback.

Response to Reviewers' Comment

Reviewer #1 (Remarks to the Author):

The authors have satisfactorily addressed most of my concerns. However, the term “three birds with one stone”, while catchy, may still be somewhat overstated without direct comparative efficacy data against other multi-target therapies. The authors should tone down the phrasing unless they can provide comparative evidence.

Response : We sincerely appreciate the reviewers' constructive suggestions. We have revised the manuscript title to *Organic di-selenide hydrogel microspheres for multimodal treatment of osteoarthritis* in accordance with your recommendations and the journal's guidelines. The corresponding section within the manuscript has also been amended accordingly.

Reviewer #3 (Remarks to the Author):

Comments: The authors have satisfactorily addressed the majority of my previous concerns through extensive revisions. The manuscript has been significantly improved in terms of experimental justification, data presentation, and clarity. In my view, the paper now meets the scientific standards of Nature Communications. I recommend acceptance pending minor revisions, primarily focused on further polishing of the language to enhance readability and precision.

Response: We are extremely grateful for your positive comments. We have polished the manuscript to enhance its readability.

Reviewer #4 (Remarks to the Author):

The authors have made substantial improvements in response to the previous reviewers' comments. The manuscript now presents a more complete and coherent study, with clarified methodologies and strengthened data interpretation. The conceptual framework of a ROS/MMP13 dual-responsive organic selenium hydrogel microsphere (HSPHR) for multi-tissue synergistic therapy in osteoarthritis is both innovative and well-supported by experimental evidence. The mechanistic insights into TXNRD1 as a key regulator across cartilage, synovium, and subchondral bone pathways significantly enhance the scientific value

of this work.

However, a few minor revisions are still required to meet the journal's publication standards.

Minor Comments:

1. Some subheadings are quite long, for example, "HSPHR hydrogel microspheres improve cartilage metabolic homeostasis in osteoarthritis by restoring mitochondrial function...". It is suggested to refine them appropriately to conform to the standard scientific writing format, which will facilitate readers in quickly locating the key points of the research.

Response: We express our sincere gratitude for your valuable suggestion. In response, we have refined certain excessively lengthy subheadings in the revised manuscript, thereby improving both the precision of the research and the overall readability of the document.

2. Please provide the full form when an abbreviation appears for the first time (e.g., MRP), and unify the font and line spacing.

Response: We express our gratitude for your insightful suggestion. We have meticulously reviewed all abbreviations appearing for the first time in the text to ensure they are accompanied by comprehensive explanations at their initial occurrence.

3. Please check whether the label "Glucose" on the ECAR experiment graph in the article should be replaced with "Rot/AA" and make the necessary changes.

Response: We express our sincere gratitude for your constructive suggestion and offer our deepest apologies for the inaccuracies in labeling presented in the ECAR results. This issue has been comprehensively addressed and clarified in the Methods section, and we have subsequently corrected the labeling of the experimental components of ECAR in Figure 6.

4. The discussion section is overly lengthy and should be streamlined, such as the discussion on SeNPs in Fig. 3.

Response: We sincerely appreciate your constructive suggestion. We have refined all sections of the discussion in the manuscript and streamlined the discussion in Figure 3.